# Revealing the Electrochemistry of All-Solid-State Li-SeS$_2$ Battery via Transmission Electron Microscopy

**Rui Yan** [1], **Fangchao Liu** [1,*] **and Zhengwen Fu** [2]

[1] School of Materials Engineering, Shanghai University of Engineering Science, Shanghai 201620, China; yr1012mgtc@163.com

[2] Department of Chemistry, Fudan University, Shanghai 200433, China; zwfu@fudan.edu.cn

[*] Correspondence: 05160011@sues.edu.cn

**Abstract:** Li-SeS$_2$ batteries balance the opposing and complimentary qualities of Li-S and Li-Se batteries by having a high specific capacity and high electrical conductivity. However, there is still a lack of knowledge regarding the electrochemical characteristics of Li-SeS$_2$ all-solid-state batteries (ASSB). Herein, transmission electron microscopy (TEM) is used to reveal the electrochemistry of a Li-SeS$_2$ battery. It is discovered that, without the Polyethylene glycol (PEG), amorphous SeS$_2$ in Li-SeS$_2$ ASSB change into crystalline selenium and a small amount of sulfur. The continuous loss of sulfur from the active material may be related to the failure of the cell at 15 cycles and the severe instability of the Coulombic efficiency. It was found that the PEG coating selenium disulfide graphene composite (PEG@rGO-SeS$_2$) cathode maintained a specific capacity of 258 mAh g$^{-1}$ and a stable Coulombic efficiency of about 97% after 50 cycles. TEM analysis shows that the charging product remains as a granular amorphous selenium disulfide with a constant Se/S ratio during cycling. The PEG-protected selenium disulfide can effectively limit the loss of elemental sulfur and regulate the reaction mechanism of the Li-SeS$_2$ batteries.

**Keywords:** all-solid-state; PEG-coated; mechanism





## 1. Introduction

The increasing urgency for and interest in sustainable energy sources can be attributed to many factors, including economic development, increased industrialization, urbanization, energy shortages, and environmental pollution caused by traditional energy use [1,2]. A reasonable response to environmental problems requires promoting the reduction of greenhouse gases, mainly carbon dioxide, so as to achieve carbon neutrality [3–5]. Battery technology is recyclable as well as environmentally friendly and nonpolluting, making it widely used in various fields. There has been much research into lithium-ion batteries, attributing to the advantage of high energy density and long lifetime [6–8]. However, conventional liquid lithium-ion batteries have serious safety issues and a low voltage window, which directly limit the range of cathode materials available and thus limits the development of energy density [9,10]. The volatile and flammable liquid electrolyte is the largest hidden problem of lithium-ion batteries. Compared with liquid electrolytes, solid electrolytes have more stable electrochemical properties and are compatible with highly reactive lithium metal anodes [11]. At the same time, solid-state electrolytes can inhibit the precipitation of lithium dendrites, make a large voltage window, and can hopefully solve the intrinsic safety problem. All-solid-state batteries (ASSBs) have captured vast interest [12]. Sulfur is considered a favorable cathode. The major advantages that promote the practical applications of lithium-sulfur (Li-S) batteries are their large abundance, low cost, and ultrahigh theoretical specific capacity (1673 mAh g$^{-1}$) [13–15]. However, Li-S all-solid-state batteries have significantly lower electrochemical performance due to the lower ionic and electronic conductivity of sulfur itself [16]. As selenium (Se) has chemical

properties similar to sulfur, Se is considered a potential candidate. However, this battery technology has several critical drawbacks that limit its performance directly caused by the use of elemental Se, such as low gravimetric specific capacity (675 mAh $g^{-1}$) and the high cost of selenium [17,18].

To reasonably balance the complementary properties of Se and S, researchers have investigated the potential of S/Se solid solution ($Se_xS_y$) in the carbon matrix, and demonstrated that it provides a strong performance gain [19,20]. $Se_xS_y$ can not only deliver a higher specific capacity and energy density than Se, but also provide improved conductivity and enhanced reaction kinetics compared with S. Inspired by this observation, $Se_xS_y$ have captured vast interest. In 2012, Amine et al. first proposed selenium sulfide ($SeS_2$) as a cathode material for lithium-ion batteries, and thoroughly studied its electrochemical reaction mechanism. With the increase of the stoichiometry of sulfur element in selenium sulfur compounds, the specific capacity of sulfur element will be improved, but its cycle stability will be worse. After weighing factors such as specific capacity, cycle life, and Coulombic efficiency, $SeS_2$ (1125 mAh $g^{-1}$) has been proved repeatedly as an excellent cathode material [21]. In 2019, Sun et al. reported for the first time the way to achieve high utilization of a S cathode by introducing Se to form a selenium-sulfur solid solution in solid-state batteries. In a series of solid-state batteries with different selenium-sulfur compounds, the $SeS_2/Li_{10}GeP_2S_{12}$-$Li_3PS_4$/Li solid-state battery discharged a specific capacity of more than 1100 mAh $g^{-1}$ at 50 mA $g^{-1}$, reaching 98.5% of the theoretical capacity. It was found that $SeS_2$ cathode exhibited a discharge-specific capacity 887 mAh $g^{-1}$ at 1 A $g^{-1}$, indicating high rate performance, and the capacity remained stable after 100 cycles [22]. Many methods have been developed to improve the electrochemical performance of $SeS_2$, such as selecting suitable conductive carbon and doping carbon materials to encapsulate $SeS_2$ [23–26]. However, it is worth nothing that the charging and discharging mechanism of the $Se_xS_y$ cathode and the capacity decay mechanism of the all-solid-state battery have rarely been studied. Wang et al. used in situ TEM to reveal the electrochemical reaction mechanism of a Li-$SeS_2$ all-solid-state battery with an electrolyte of $Li_{10}Si_{0.3}PS_{6.7}Cl_{1.8}$ (LSP-SCl); during initial lithiation, $SeS_2$ was converted to nanocrystalline $Li_2Se$ dispersed in a $Li_nSeS_2$ matrix. Then, after additional lithiation, crystalline $Li_2S$ was produced, and high-resolution electron microscopy (HRTEM) revealed that the crystalline $Li_2Se$ and $Li_2S$ were phase-separated. Polycrystalline $Li_2Se$ was transformed to Se during the charging process, whereas $Li_2S$ was converted to S, indicating that the result after charging was a mixture of Se and S [27]. Moreover, the detailed mechanism of $SeS_2$-based cathodes in solid electrolytes is still unknown. Therefore, understanding the mechanistic processes of the battery is the key to the successful development of high-performance lithium batteries [28–30].

Here, the synthetic rGO-$SeS_2$ material and the PEG@rGO-$SeS_2$ material's mechanisms have been put to the test. The mechanism of the rGO-$SeS_2$ electrode was investigated using TEM tests, and it was found that the amorphous selenium sulfide on the graphene surface transformed to crystalline selenium and a small amount of sulfur as the charge and discharge proceeded. The failure process occurring in the batteries and the instability of the Coulombic efficiency may be related to the continuous loss of sulfur from the active material. Additionally, the obtained composite electrode sheet PEG@rGO-$SeS_2$ electrode was studied electrochemically. The first discharge-specific capacity of PEG@rGO-$SeS_2$ electrode was 1124.7 mAh $g^{-1}$, and the specific capacity after 50 cycles was 258.7 mAh $g^{-1}$. TEM tests showed that the charging product remained as granular amorphous selenium disulfide after multiple cycles, while the selenium-sulfur ratio remained unchanged in the EDS energy spectrum data. Therefore, elemental sulfur in the electrode was prevented from entering the electrolyte by PEG coating on rGO-$SeS_2$. The use of PEG for the protection of selenium disulfide can effectively suppress the loss of sulfur element and regulate the reaction mechanism of Li-$SeS_2$ batteries.

## 2. Results and Discussion

The XRD characterization was used to test the rGO-SeS$_2$ synthesized using the in situ synthesis method, whose active substance selenium disulfide is in the amorphous form, and the XRD test results confirmed, as shown in Figure 1a. The XRD diffraction peak of rGO-SeS$_2$ is the same as the peak of rGO, and there is no characteristic diffraction peak of selenium disulfide at the peak of 2θ = 24° corresponding to the (022) crystallographic plane diffraction of graphene [31], indicating that the selenium disulfide on graphene is in the amorphous form [32,33]. From the thermogravimetric analysis (TGA) curves in Figure 1b, it can be seen that the rGO-SeS$_2$ material has essentially no weight loss at temperatures up to 270 °C. As the temperature increases, the weight starts to decrease, and the weight still reaches 65% retention when the temperature reaches 520 °C. Figure 1c shows the Raman spectra of rGO-SeS$_2$ materials recorded under N$_2$ atmosphere. The Raman peaks of about 1342 and 1583 cm$^{-1}$ can be assigned to the disordered graphite band (D-band) and the crystalline band (G-band) with a ratio of ID/IG = 1.34, indicating the degree of graphitization of the carbon.

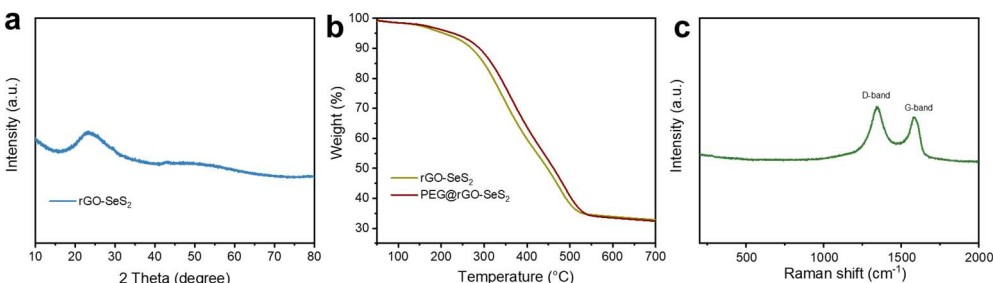

**Figure 1.** (**a**) XRD analysis of rGO-SeS$_2$ materials. (**b**) Thermogram of rGO-SeS$_2$ and PEG@rGO-SeS$_2$ materials under N$_2$ atmosphere. (**c**) Raman spectra of rGO-SeS$_2$ materials.

From Figure 2a, SEM images recorded clearly show the three-dimensional network cross-linked structure shape of the graphene without obvious aggregation of SeS$_2$ particles. The result of the back-scattering electron (BSE) was only found in the structure of graphene, as shown in Figure 2b. This may be due to the high dispersion of selenium disulfide on the graphene surface. Figure 2d shows the EDS mapping of rGO-SeS$_2$. What is clear about the figure is uniform Se and S on the surface of rGO-SeS$_2$, which is consistent with the results of BSE. TEM is used to characterize the structure of rGO-SeS$_2$ materials. Figure 2d shows only the graphene structure, which is a three-dimensional mesh structure, with no particles attached to graphene and no diffracted spots in the selected area electron diffraction (SAED), proving that the rGO-SeS$_2$ material is amorphous, which is consistent with the XRD results.

The electrochemical performance and charging mechanism of solid-state Li-SeS$_2$ batteries were studied using rGO-SeS$_2$ as the cathode. Figure 3a shows the rate performance test of rGO-SeS$_2$ material. When the current density was C/32, C/16, and C/8, the initial discharge-specific capacity reached 1130, 1129, and 1123 mAh g$^{-1}$, which reached the theoretical capacity of selenium disulfide. When the current density was C/4, C/2, and 1C, the discharge-specific capacity was 103, 902, and 773 mAh g$^{-1}$, respectively. It was found that, when the current density increased to 1C, the discharge capacity was only 68% of the theoretical capacity. Figure 3b shows the initial dozen cycles of charging and discharging at a current density of C/8 for voltages from 1.6 V to 2.6 V and the Coulombic efficiency. Failure process occurred when the battery was cycled to the 15th turn at C/8 current density and the Coulombic efficiency was unstable.

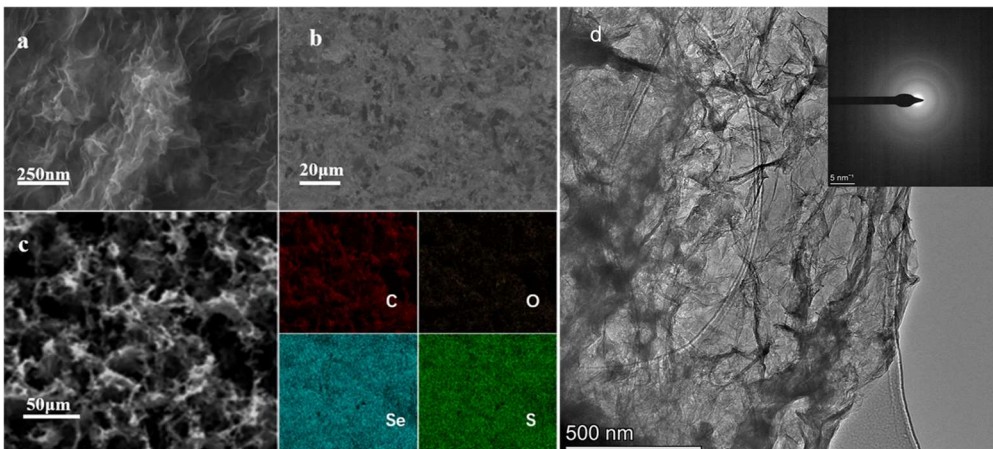

**Figure 2.** (**a**) SEM image of rGO-SeS$_2$. (**b**) BSE image of rGO-SeS$_2$ material. (**c**) EDS analysis of rGO-SeS$_2$ electrode. (**d**) TEM and SAED images of rGO-SeS$_2$ material.

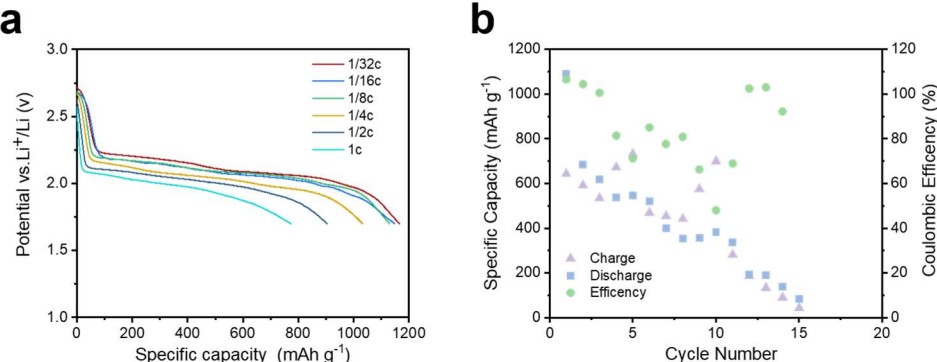

**Figure 3.** At 50 °C, (**a**) rate performance of rGOSeS$_2$ cathode; (**b**) cycling performance and Coulombic efficiency at C/8 current density.

The mechanism of the rGO-SeS$_2$ electrode was investigated using TEM and SAED. Figure 4a shows the TEM image after charging for the first time at C/8 current density; it was found that the highly dispersed amorphous selenium disulfide became particles loaded onto the graphene around 40 nm during charging, so the selenium and sulfur elements were aggregated to form particles during the first charging process. It was also observed that the SAED diffractogram had no diffraction spot and only had the diffraction characteristics of graphene, which indicates that the particles generated by charging were amorphous at this time. With the increase in charging times, the mechanism was investigated when the charging times reached the fifth time. Figure 4b shows the TEM image after the fifth time charging at C/8 current density, and it can be observed that the particles around 150 nm are distributed on the graphene surface, indicating that the selenium and sulfur elements were continuously aggregated as the cycle continued. It was also observed that the SAED diffractogram had no diffracted spots and only had the diffraction characteristics of graphene, indicating that the particles generated by the fifth charge were also amorphous. Figure 4c shows the TEM image after the tenth charge, and it can be seen that there were particles around 100 nm distributed on the graphene surface, indicating that no agglomeration of Se and S elements occurred at this time. It was also observed that there were diffracted spots in the SAED diffractogram with d spacing = 0.287, 0.165, and 0.202 nm corresponding to the (211), (042), and (202) crystalline planes of Se (PDF #32-0992; Se), indicating that the particles generated by charging at this time were crystalline selenium. Figure 4d shows the mechanism after the thirteenth charge, and it can be seen that particles around 600 nm were distributed on the surface of graphene, indicating that there was a continuous enlargement of the particles. It was also observed that there were diffracted spots in the SAED diffractogram when the atomic spacing was

equal to 0.143, 0.191 nm corresponding to the (113), (122) crystalline planes of selenium (PDF #32-0992; Se), respectively, suggesting that the particles generated by charging at this time were crystalline selenium. As shown in Figure 4e,f, the EDS distribution diagram shows that the particles are composed of selenium only and the weight ratio of selenium is 30.87% and the weight ratio of S is 2.22%, indicating that the elemental sulfur was continuously lost during the cycling process.

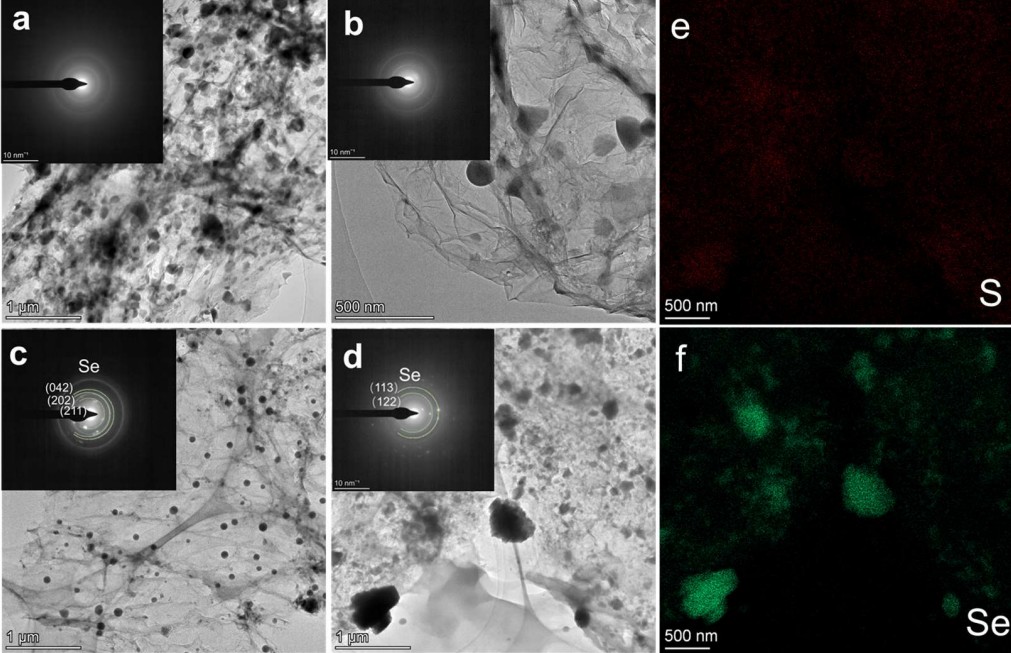

**Figure 4.** SAED and TEM images of rGO-amor $SeS_2$; (**a**) first charge; (**b**) fifth charge; (**c**) tenth charge; (**d**) thirteenth charge; (**e,f**) EDS distribution of S and Se at the thirteenth charge.

The mechanism of the rGO-$SeS_2$ electrode was investigated using TEM, SAED, and EDS tests, and it was found that the amorphous selenium sulfide on the graphene surface transforms to crystalline singlet selenium and a small amount of sulfur as charging and discharging proceeds. The cell capacity decay and Coulombic efficiency instability may be related to the continuous loss of active material sulfur. Therefore, the modulation of the cell mechanism was achieved by PEG-coating the rGO-$SeS_2$ electrode to improve the cycle stability of Li-$SeS_2$ batteries. A cyclic performance test was conducted in a vacuum oven at 50 °C. Figure 5a shows the performance of PEG@rGO-$SeS_2$ positive electrode cycling for 50 times at C/8 current density. The initial discharge-specific capacity of PEG@rGO-$SeS_2$ positive electrode was 1124 mAh $g^{-1}$, reaching the theoretical capacity of $SeS_2$, indicating that this concentration of PEG-coated rGO-crys $SeS_2$ electrode did not impede the first ion transport. The discharge-specific capacity reached 258 mAh $g^{-1}$ after 50 cycles. Although the Coulombic efficiency was low for the first time, it stabilized at about 97% with the increase in the number of cycles. Therefore, coating the surface of rGO-$SeS_2$ with PEG could improve the number of cycles and stabilize the Coulombic efficiency of the battery. PEG acts as a barrier to limit the polysulfide to the positive electrode, thus reducing the mass loss of the active substance, which was verified in the following mechanism tests. As Figure 5b shows the TEM image after the fifth charge at C/8 current density, it can be seen that there are particles distributed on the surface of the graphene. Meanwhile, there is no diffraction spot in the SAED diffractogram, and only the diffraction ring of graphene can be observed, indicating that the particles formed at this time are in an amorphous state. As the number of cycles increased to fifty, scanning transmission electron microscopy images and selected electron diffraction revealed amorphous particles loaded on the graphene from Figure 5c. The results show that the active substances coated with PEG exist in amorphous form during the charge-discharge cycle. Moreover, we can see from Figure 5d–f that the

particles are composed of selenium and sulfur. The selenium to sulfur ratio is still the same as it was on the first charge, indicating that there is no loss of active substance S at this time, which once again proves that PEG can prevent the loss of active substance in the all-solid-state Li-SeS$_2$ battery. The PEG with a linear structure is wrapped around the selenium disulfide, which plays a role in protecting the selenium disulfide and effectively prevents the generated Li$_2$S and Li$_2$Se from entering the electrolyte, thus preventing the loss of elemental sulfur.

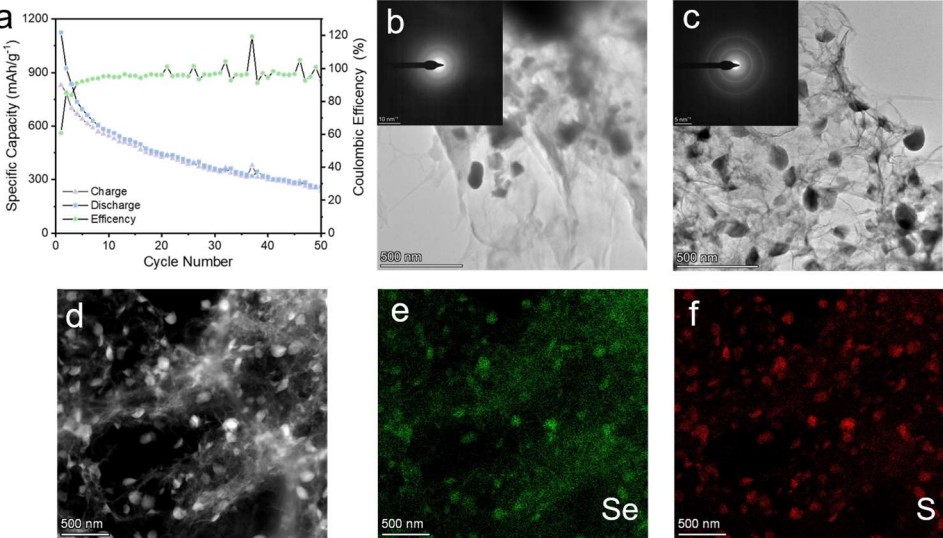

**Figure 5.** (**a**) SAED cycling performance and Coulombic efficiency of PEG@rGO-SeS$_2$cathode at 1/8C current density; SAED and TEM images of rGO-SeS$_2$: (**b**) fifth charge; (**c**) fiftieth charge; (**d**–**f**) EDS distribution of S and Se at the fifth charge.

## 3. Materials and Methods

### 3.1. Materials

Graphite was purchased from Sinopharm Chemical Reagent Co., Ltd. (Shanghai, China). Nitrate of potash, selenium powder, sublimed sulfur, lithium iodide, concentrated sulfuric acid, and potassium permanganate were purchased from Shanghai Titan Co., Ltd. (Shanghai, China). Sodium sulfide nonahydrate, hydroiodic acid, hydrogen peroxide, and lithium carbonate tablets were purchased from Afa Aisa Chemical Co., Ltd. (Shanghai, China). Three-hydroxypropionitrile, polyethylene glycol (PEG) were purchased from Aladdin Reagents Co., Ltd. (Shanghai, China). Ultrapure water (18.2 MΩ cm) was used in all of the experiments.

### 3.2. Synthesis of rGO-SeS$_2$

The dispersion liquid, graphene oxide (GO), was synthesized through Hummers' method; 0.1047 g Na$_2$S·9H$_2$O, 0.139 g elemental sulfur, and 0.349 g elemental selenium were dissolved in a 20 mL deionized water. Then, the mixed solution was stirred at room temperature for 6 h to obtain a uniform and transparent burgundy Na$_2$SeS$_2$ solution. Typically, the rGO-SeS$_2$ were obtained via directly absorb 2.8 mL 3 mg/mL of GO dispersion and 1.1 mL 0.48 mmol/2.2 mL of Na$_2$SeS$_2$ solution were mixed under acidic conditions and heated at 110 °C for 75 min. The resulting rGO-SeS$_2$ were collected by being washed with deionized water several times and put into a reaction kettle with deoxygenated deionized water as a solution. Finally, after naturally cooling down, the rGO-SeS$_2$ composite was obtained after being freeze-dried for 24 h.

### 3.3. Synthesis of PEG@rGO-SeS$_2$

PEG@rGO-SeS$_2$ was synthesized as shown in Figure 6. Typically, the rGO-SeS$_2$ were obtained via directly absorbing 2.8 mL 3 mg/mL of GO dispersion, and 1.1 mL 0.48 mmol/2.2 mL

of $Na_2SeS_2$ solution were mixed under acidic conditions and heated at 95 °C for 75 min. The synthesized $rGO$-$SeS_2$ electrode sheets were immersed in 1.8 g/20 mL concentration of PEG solution for a certain time and the obtained materials were freeze-dried to obtain PEG@$rGO$-$SeS_2$ electrode sheets.

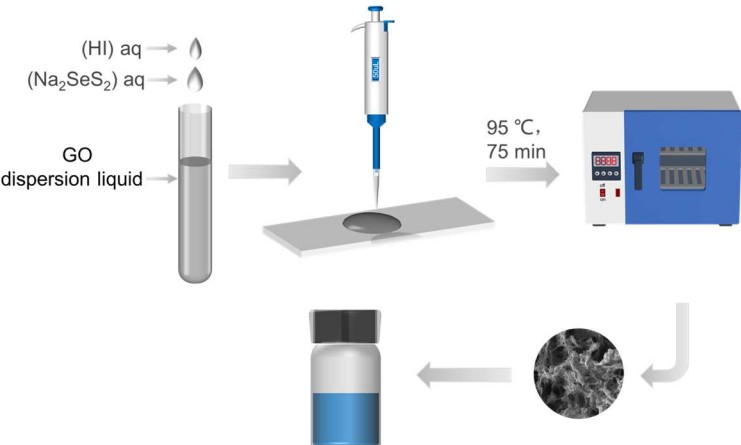

**Figure 6.** Schematic diagram of the PEG@$rGO$-$SeS_2$ preparation process.

### 3.4. Synthesis of Solid Electrolyte LiIHPN-0.5LiI (SMC)

The solid electrolyte LiIHPN-0.5LiI (SMC) was prepared by mixing 3-hydroxypropionitrile (HPN) and high-purity lithium iodide crystals (LiI) with a molar ratio of 2:3 at room temperature followed by vigorous magnetic stirring, which was conducted in an anhydrous and dry glove box. The mixture was heated and stirred for 24 h at 120 °C, leading to the formation of homogeneous liquid. At normal temperature, the produced small molecule composite electrolyte has a conductivity of $1.25 \times 10^{-6}$ S cm$^{-1}$, and it is around 25 μm thick. Low-melting-point electrolytes, as opposed to high-melting-point inorganic solid electrolytes, can permeate the composite by making strong contact with the electrode sheet after melting.

### 3.5. Characterizations

The microstructure of materials, revealed by using a scanning electron microscope (SEM, VEGA3 TESCAN), are well-observed from low magnification to high magnification. Transmission electron microscopy (TEM, JEOL 2100F) and selected area electron diffraction (SAED) were used to study the morphology and microstructure of the electrodes after charging. The powder X-ray diffraction (XRD) were carried out at a scanning rate of $0.4°$ s$^{-1}$ in the 2θ range from 5° to 100° with Cu Kα radiation. The determinations of $SeS_2$ were analyzed by using thermogravimetry (TG) with a heating rate of 10 °C/min under $N_2$ atmosphere. The Raman spectrum was collected on a Raman spectrometer (Horiba Jobin Yvon LabRAM HR800).

### 3.6. Electrochemical Measurements

The electrochemical performance of the $rGO$-$SeS_2$ cathode was evaluated using an all-solid-state battery containing the lithium plate (purity 99.999%) as an anode and a single-layer paper towel (Vinda) as a separator. The all-solid-state batteries were assembled in a glove box filled with argon gas. To assemble the battery, the prepared solid electrolyte LiIHPN-0.5LiI was heated in an oil bath and converted into a liquid with good fluidity, which infiltrated the composite electrode material and was placed on a separator and permeated for 30 min. Subsequently, a lithium plate was pressed onto an electrode material filled with electrolyte. The galvanostatic charge–discharge of batteries was tested with a Land CT-2001A battery test system (Wuhan, China) at different current densities.

## 4. Conclusions

TEM was used to elucidate the electrochemical reaction process of a Li-$SeS_2$ ASSB. The Coulombic efficiency of the $rGO$-$SeS_2$ synthesized in situ used as the battery's cathode was

unstable and quickly failed throughout 15 charge and discharge cycles. It was discovered that the charging and discharging process caused the $SeS_2$ amorphization on the surface of the graphene to change into crystalline elemental selenium and a small quantity of sulfur. The ongoing loss of the active ingredient sulfur may be responsible for the cell capacity attenuation and Coulombic efficiency instability. In contrast, the PEG@rGO-crys $SeS_2$ electrode still had some capacity retention and stable Coulombic efficiency after 50 cycles. TEM examination revealed that during cycling, the charge product remained as granular amorphous selenium disulfide with a constant Se/S ratio. Selenium disulfide can be effectively protected by PEG to stop the loss of elemental sulfur. Therefore, the PEG@rGO-crys $SeS_2$ electrode obtained by PEG-coating rGO-$SeS_2$ can be used to tune the cell mechanism. Our research provides new insights into the electrochemistry of Li-$SeS_2$ ASSB, which is very important for optimization of Li-$SeS_2$ battery systems.

**Author Contributions:** Methodology, data curation, writing—original draft preparation, writing—review and editing, R.Y.; investigation, resources, funding acquisition, F.L.; supervision, project administration, Z.F. All authors have read and agreed to the published version of the manuscript.

**Funding:** The National Natural Science Foundation of China (NO.2170031075).

**Data Availability Statement:** All data that support the findings of this study are included within the article.

**Acknowledgments:** Class III Peak Discipline of Shanghai—Materials Science and Engineering (High-Energy Beam Intelligent Processing and Green Manufacturing).

**Conflicts of Interest:** The authors declare no conflict of interest.

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
