# Peer review of "Revealing the Electrochemistry of All-Solid-State Li-SeS2 Battery via Transmission Electron Microscopy"

_inorganics, doi:10.3390/inorganics11060257_

Round 1
Reviewer 1 Report
This article reports that the mechanism of the all-solid-state Li-SeS2 battery was studied by transmission electron microscopy, and the tuning of the battery mechanism was realized to improve the cycle stability of the Li-SeS2 battery. It is worth affirming that the author has rigorous logic and excellent drawing skills. However, there are still some details in the article that are to be strengthened. In my opinion, this manuscript can be accepted after the following questions are addressed.
1. The language of the abstract is not concise enough and does not express the ideas of the paper concisely, and should be revised appropriately.
2. In the introduction, recent progress on sulfur cathodes is suggested to be cited: Advanced Materials, 2019, 31, 1901220; Electrochemical Energy Reviews,2018, 1, 239–293; Chemical Society Reviews,2020, 49, 2140-2195; Advanced Functional Materials, 2019, 29, 1806724.
3. what was the spectral resolution of Raman spectroscopy? Please reinterpret Figure 1c.
4. Attention to detail should be paid to drawing the diagrams. For example, the TEM image in Figure 2 is not aligned with the left image.
5. The image in Figure 5a is not clear, it is recommended to provide a high-resolution image.
6. The author should explain in detail how this work addresses the loss of sulfur elements.
Author Response
1.The language of the abstract is not concise enough and does not express the ideas of the paper concisely, and should be revised appropriately.
I have condensed the abstract, which can be seen in lines 8-21 of the article.
2.In the introduction, recent progress on sulfur cathodes is suggested to be cited: Advanced Materials, 2019, 31, 1901220; Electrochemical Energy Reviews,2018, 1, 239–293; Chemical Society Reviews,2020, 49, 2140-2195; Advanced Functional Materials, 2019, 29, 1806724.
I have cited the above articles in the order of 13-15.
3. what was the spectral resolution of Raman spectroscopy? Please reinterpret Figure 1c.
I have reinterpreted Figure 1c, which can be seen in lines 111-114 of the article
4.Attention to detail should be paid to drawing the diagrams. For example, the TEM image in Figure 2 is not aligned with the left image.
I have made changes, as shown in Figure 2.
5.The image in Figure 5a is not clear, it is recommended to provide a high-resolution image.
Replacement has been performed, as shown in Figure 5a.
6.The author should explain in detail how this work addresses the loss of sulfur elements.
A detailed explanation can be seen in lines 207-210 of the article.
Reviewer 2 Report
The authors show the electrochemical characteristics of Li-SeS2 batteries and the benefits of PEG-coated rGO-SeS2 cathodes in preserving specific capacity and ensuring stability in Coulomb efficiency over multiple cycles. These findings contribute to the advancement and understanding of Li-SeS2 battery systems, fostering potential improvements in their design and performance.
I would suggest the authors improve their manuscript considering the following aspects.
1. Usually in solid-state batteries, solid electrolytes are rate-determining components. I would strongly suggest including the electrochemical characterization of solid electrolytes in the main manuscript.
2. Solid-state Li-SeS2 batteries is one of a nice idea, at the same time it would be nice to include some of the other studies related to Li-SeS2 batteries!
Author Response
1. Usually in solid-state batteries, solid electrolytes are rate-determining components. I would strongly suggest including the electrochemical characterization of solid electrolytes in the main manuscript.
A detailed explanation of electrolytes can be observed in lines 246-250 of the article.
2. Solid-state Li-SeS2 batteries is one of a nice idea, at the same time it would be nice to include some of the other studies related to Li-SeS2 batteries!
Additional studies of all-solid-state li-SeS2 cells can be observed in lines 56-79 of the article.